# VARIATIONAL QUANTIZATION FOR STATE SPACE MODELS

## ABSTRACT

Forecasting tasks using large datasets gathering thousands of heterogeneous time series is a crucial statistical problem in numerous sectors. The main challenge is to model a rich variety of time series, leverage any available external signals and provide sharp predictions with statistical guarantees. In this work, we propose a new forecasting model that combines discrete state space hidden Markov models with recent neural network architectures. We introduce a variational discrete posterior distribution of the latent states given the observations and a two-stage training procedure to alternatively train the parameters of the latent states and of the emission distributions. By learning a collection of emission laws and temporarily activating them depending on the hidden process dynamics, the proposed method allows exploring large datasets, exploiting available external signals and providing probabilistic predictions. We assess the performance of the proposed method using several datasets and show that it outperforms other state-of-the-art solutions.

## 1 INTRODUCTION

An increasingly common time series forecasting problem concerns the forecast of large datasets gathering thousands of heterogeneous sequences, see Makridakis et al. (2018; 2022); Lai et al. (2018); Zhou et al. (2021b); David et al. (2022a) and the references therein. One of the main difficulties is to design mathematical models for a large variety of seasonal patterns, noise levels, trends and non-stationary changes. Additionally, some time-series datasets provide external signals that can be exploited to detect behaviors in the main time series that would otherwise be missed (David et al., 2022a;b). Regarding this new type of forecasting use case, state-of-the-art solutions do not provide satisfactory results yet.

Parametric statistical models have been largely studied during the past decades, see for instance Box et al. (2015); Hyndman & Athanasopoulos (2018). Based on a sharp modeling of the time series distribution, these models can compute accurate predictions along with confidence intervals that make them largely used in numerous applications. Depending on the nature of the use case, many approaches have been proposed. The exponential smoothing model (Brown & Meyer, 1961), the Trigonometric Box-Cox transform, ARMA errors, Trend, and Seasonal components model (TBATS) (Livera et al., 2011), or the ARIMA model with the Box-Jenkins approach (Box et al., 2015) are for instance very popular parametric generative models. However, they cannot be used for large datasets gathering thousands of time series. As a new model needs to be trained for each new time series, the training process can take considerable time depending on the number of sequences. Furthermore, much of the parametric models proposed cannot include external signals in their framework as the exact dependencies between the additional signals and the main ones remain unknown.

Hidden Markov models are other widespread models that have been largely studied in the literature (Särkkä, 2013; Douc et al., 2014; Chopin et al., 2020). Introduced in the late 1960s, these generative models rely on hidden processes to describe the distribution of the target time series. Numerous variations have been proposed to fit different use cases (Juang & Rabiner, 1985; Douc et al., 2004; Touron, 2019). In addition to providing accurate predictions, these models are supported by solid theoretical results on their identifiability and their consistency, see for instance Douc et al. (2011); Gassiat & Rousseau (2016); Gassiat et al. (2020) and references therein. However, when large datasets are considered, as a hidden state model has to be trained on each new time series, they are

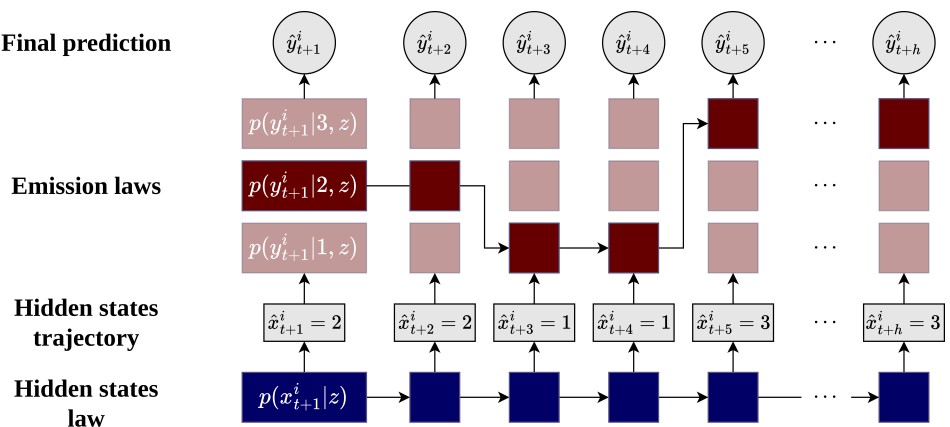

Figure 1: Illustration of the proposed framework with 3 hidden states. Given the past of a time series $y^i_{t-w+1:t}$ and possible additional external signals $w^i_{t-w+1:t}$ (called $z$ in the figure), a trajectory of the hidden state $\hat{x}^i_{t+1:t+h}$ is drawn using the law of the hidden states. Then, conditionally to the values taken by the hidden process, one of the emission laws is activated and used to compute the final prediction $\hat{y}^i_{t+1:t+h}$. The trajectory displayed for the hidden state is arbitrary and provided for illustration purposes.

not well suited to forecast large samples gathering thousands of time series. Nevertheless, several contributions introducing hidden Markov models able to leverage external signals have been proposed, see Bengio & Frasconi (1994); Radenen & Artieres (2012); Gonzalez et al. (2005); David et al. (2022b).

Finally, with recent improvements in speech processing and image recognition, neural-network-based models have emerged as the new state-of-the-art in time series forecasting. Among them, recurrent neural networks or sequence to sequence deep learning architectures (Hochreiter & Schmidhuber, 1997; Vaswani et al., 2017) offer very appealing alternatives to exploit large time series dataset and leverage any kind of external signals. The DeepAR methods (Salinas et al., 2020), N-HiTS and N-BEATS frameworks (Oreshkin et al., 2019; Challu et al., 2023) and the following Transformer-based approaches Lim et al. (2021); Zhou et al. (2021b; 2022); Woo et al. (2022); Wu et al. (2022); Liu et al. (2022); Woo et al. (2023); Wu et al. (2023); Nie et al. (2023) are examples of neural-network-based models that have obtained unprecedented accuracy levels in various applications. However, predictions computed by these methods are not interpretable and only a handful of theoretical results have been provided with these architectures.

In this paper, we introduce a new forecasting method combining hidden Markov models with recent neural-networks-based models. In this framework, it is assumed that time series are ruled by hidden Markov processes modeling the internal state of the time series. Depending on the hidden states dynamics, several emission laws are learned and specialized at forecasting specific types of behaviours. Maximum likelihood approaches cannot be used directly to train such a model and Expectation-Maximization (EM) algorithm is commonly used in this case. However, this algorithm is computationally costly, requires a fair amount of tuning and is very sensitive to the initialization. Thus, inspired by ideas brought with the Vector quantized VAE model (van den Oord et al., 2018), a training process based on the Evidence Lower BOund (ELBO) learning alternatively the parameters of the emission distributions and of the latent states is introduced. On a collection of reference datasets, our approach outperforms current state-of-the-art solutions while providing probabilistic predictions. Furthermore, we show that the model can forecast non-stationary time series, especially when relevant external signals are included in the hidden states and the emission distributions.

The paper is organized as follows. The proposed model is presented in Section 2 along with the training procedure. Then, a complete experimental study is provided in Section 3 where the proposed framework is applied on several datasets, its accuracy assessed and evaluated in comparison with a collection of other state-of-the-art methods. Finally, some research perspectives are given in Section 4.

## 2 MODEL AND TRAINING PROCEDURE

### 2.1 MODEL FORMULATION

Consider a dataset gathering $N \in \mathbb{N}^*$ time series. For $i \in \{1, \cdots, N\}$, let $(y_t^i)_{t \in \mathbb{Z}}$ be the observation of the sequence $i$ and $(w_t^i)_{t \in \mathbb{Z}}$ a sequence of additional signals. These auxiliary variables may account for the history of some additional time series, or any other available information. The aim of the proposed model is to forecast, for all $i \in \{1, \cdots, N\}$, the next $h \geq 1$ values of $y^i$ based on the past $w \geq 1$ values of $y^i$ and $w^i$, i.e. to estimate $p_\theta(y_{t+1:t+h}^i | y_{t-w+1:t}^i, w_{t-w+1:t}^i)$ the probability density function of the time series when the parameter value is $\theta$. We assume the existence of an additional discrete hidden process denoted by $(x_t^i)_{T+1 \leq t \leq T+h}$ taking value in $\mathsf{X} = \{1, \cdots, K\}$ and that rules the density of $y_{t+1:t+h}^i$. This discrete hidden signal can be interpreted as a state or a regime in which is a sequence $i$ is at a time t. Depending on the values taken, $K$ different predictions can be computed for a same time series, all representing behaviours linked to the hidden regime of the time series. Thus, the previous density can be written as follows:

$$
p_\theta(y_{t+1:t+h}^i | y_{t-w+1:t}^i, w_{t-w+1:t}^i)
$$
$$
= \sum_{x_{t+1:t+h}^i \in \mathsf{X}^h} p_\theta(y_{t+1:t+h}^i, x_{t+1:t+h}^i | y_{t-w+1:t}^i, w_{t-w+1:t}^i)
$$
$$
= \sum_{x_{t+1:t+h}^i \in \mathsf{X}^h} \prod_{s=1}^{h} p_{\theta_y}(y_{t+s}^i | x_{t+1:t+s}^i, y_{t-w+1:t+s-1}^i, w_{t-w+1:t}^i)
$$
$$
\times p_{\theta_x}(x_{t+s}^i | x_{t+1:t+s-1}^i, y_{t-w+1:t+s-1}^i, w_{t-w+1:t}^i) \, ,
$$

with the convention $p_\theta(.|x_{t+1:t+s-1}^i, y_{t-w+1:t+s-1}^i, w_{t-w+1:t}^i) = p_\theta(.|y_{t-w+1:t}^i, w_{t-w+1:t}^i)$ for $s = 1$. Note that we decomposed the unknown parameter $\theta = (\theta_x, \theta_y)$ with i) the parameters corresponding to the distribution of the hidden states denoted by $\theta_x$ and ii) the parameters corresponding to the distribution of the main signal conditionally to the hidden states denoted by $\theta_y$. We consider the following assumptions.

- For all $i \in \{1, \cdots, H\}$ and all $s \in \{1, \cdots, h\}$, we assume that the conditional distribution of $y_{t+s}^i$ depends on the current value of the external signal $x_{t+s}^i$ and the window $(y_{t-w+1:t}^i, w_{t-w+1:t}^i)$.
- For all $i \in \{1, \cdots, H\}$ and all $s \in \{1, \cdots, h\}$, we assume that the conditional distribution of $x_{t+s}^i$ depends on the previous $x_{t+s-1}^i$ and the window $(y_{t-w+1:t}^i, w_{t-w+1:t}^i)$.

Thus, the predictive distribution can be written as follows.

$$
p_\theta(y_{t+1:t+h}^i | y_{t-w+1:t}^i, w_{t-w+1:t}^i)
$$
$$
= \sum_{x_{t+1:t+h}^i \in \mathsf{X}^h} \prod_{s=1}^{h} p_{\theta_y}(y_{t+s}^i | x_{t+s}^i, y_{t-w+1:t}^i, w_{t-w+1:t}^i) p_{\theta_x}(x_{t+s}^i | x_{t+s-1}^i, y_{t-w+1:t}^i, w_{t-w+1:t}^i) \, .
$$

$$(1)$$

The proposed framework is therefore a generative model composed of two parts: the distribution of the hidden process and the conditional emission distributions of the main signal. An illustration of the proposed model is presented in Figure 1.

### 2.2 TRAINING

As $(x_t^i)_{t \in \mathbb{Z}}$ is never observed, the loglikelihood cannot be computed. In this setting, Expectation Maximization (EM)-based algorithms could for instance be used to train the model, see Dempster et al. (1977). However, the models used in this paper are inspired by recent deep architectures such as DeepAR, see Salinas et al. (2020), and training these models wtih EM-based procedures is computationally very intense. A relevant approach is to substitute the loglikelihood by the Evidence Lower BOund (ELBO). For greater clarity, we omit the dependencies to $y_{t-w+1:t}^i, w_{t-w+1:t}^i$. For

all $i \in \{1, \cdots, N\}$, note that

$$\log p_\theta(y^i_{t+1:t+h}) \geq \mathbb{E}_{x \sim q_\phi} \left[ \log \frac{p_\theta(y^i_{t+1:t+h}, x^i_{t+1:t+h})}{q_\phi(x^i_{t+1:t+h}|y^i_{t+1:t+h})} \right] \,,$$

where the right hand side term defines the ELBO and $q_\phi(x^i_{t+1:t+h}|y^i_{t+1:t+h})$ the posterior variational probability of the hidden sequence $x^i_{t+1:t+h}$ conditionally to the observed sequence $y^i_{t+1:t+h}$. Therefore, the proposed loss function $\mathcal{L}(\phi, \theta_x, \theta_y)$ used to train the model is given by

$$
\begin{aligned}
\mathcal{L}(\phi, \theta_x, \theta_y) &= \frac{1}{N} \sum_{i=1}^N L^i(\phi, \theta_x, \theta_y) \\
&= \frac{1}{N} \sum_{i=1}^N \mathbb{E}_{x \sim q_\phi} \left[ \log \frac{p_\theta(y^i_{t+1:t+h}, x^i_{t+1:t+h}|y^i_{t-w+1:t}, w^i_{t-w+1:t})}{q_\phi(x^i_{t+1:t+h}|y^i_{t-w+1:t+h}, w^i_{t-w+1:t})} \right] \\
&= \frac{1}{N} \sum_{i=1}^N \left\{ \mathcal{L}^i_1(\phi, \theta_y) - \mathcal{L}^i_2(\phi) + \mathcal{L}^i_3(\phi, \theta_x) \right\} \,,
\end{aligned}
$$

with

$$
\begin{aligned}
\mathcal{L}^i_1(\phi, \theta_y) &= \mathbb{E}_{x \sim q_\phi} \left[ \sum_{s=1}^h \log p_{\theta_y}(y^i_{t+s}|x^i_{t+s}, y^i_{t-w+1:t}, w^i_{t-w+1:t}) \right] \\
\mathcal{L}^i_2(\phi) &= \mathbb{E}_{x \sim q_\phi} \left[ \sum_{s=1}^h \log q_\phi(x^i_{t+s}|y^i_{t-w+1:t+h}, w^i_{t-w+1:t}) \right] \\
\mathcal{L}^i_3(\phi, \theta_x) &= \mathbb{E}_{x \sim q_\phi} \left[ \sum_{s=1}^h \log p_{\theta_x}(x^i_{t+s}|x^i_{t+s-1}, y^i_{t-w+1:t}, w^i_{t-w+1:t}) \right] \,.
\end{aligned}
$$

**Two-step training.**     Inspired by ideas brought by the Vector quantized VAE model (van den Oord et al., 2018), the ELBO loss is decomposed into three components that are used to train the unknown distributions. The two first terms correspond to a reconstruction loss and are used to train jointly the decoder and the encoder. The last term $\theta_x \mapsto \mathcal{L}^i_3(\phi, \theta_x)$ is used to train the prior distribution , i.e. the distribution of the discrete latent states.

Following (van den Oord et al., 2018), the prior distribution is first initialized as a uniform distribution i.e. $p_{\theta_x}(x^i_{t+s}|x^i_{t+s-1}, y^i_{t-w+1:t}, w^i_{t-w+1:t})$, $1 \leq s \leq h$, is a uniform distribution and only $\phi$ and $\theta_y$ are trained by optimizing $\mathcal{L}^i_1(\phi, \theta_y)$ and $\mathcal{L}^i_2(\phi)$. This means that only the emission distributions and the posterior variational distributions are first trained. This allows to use $y^i_{t-w+1:t}, w^i_{t-w+1:t}$ as inputs which are passed through deep encoder architectures $f_{\theta_y}$ and $f_\phi$ which can be trained to produce outputs which are used to design the emission distributions and the posterior variational distributions distributions. After the convergence of the emission distributions, they are frozen and the prior model is trained, guided by the learned posterior variational distribution. This allows to train an arbitrarily complex categorical prior distribution by only optimizing $\theta_x \mapsto \mathcal{L}^i_3(\phi, \theta_x)$. See Appendix A for ablation studies on the proposed training process as well as some choices made in the model formulation and implementation.

## 2.3 Implementation

For all $i \in \{1, \cdots, N\}$ and $s \in \{1, \cdots, h\}$ consider the following assumptions.

- Inspired by the DeepAR method introduced in Salinas et al. (2020), we assume that the $K$ emission distributions $(p_{\theta_y}(y^i_{t+s}|x^i_{t+s} = k, y^i_{t-w+1:t}, w^i_{t-w+1:t}))_{1 \leq k \leq K}$ are Gaussian distributions parameterized by $K$ different neural networks components. In fact, for each hidden state $k \in \{1, \cdots, K\}$, a neural-network-based model denoted $f^k_{\theta_y}$ is trained to predict $h$ couples of parameters for the Gaussian emission distributions linked to the hidden

regime $k$: $((\mu_{t+s}^{k,i}, \sigma_{t+s}^{k,i}))_{1 \le s \le h}$ with $\mu$ the mean and $\sigma$ the standard deviation.

$$p_{\theta_y}(y_{t+s}^i | x_{t+s}^i = k, y_{t-w+1:t}^i, w_{t-w+1:t}^i) = \mathcal{N}(y_{t+s}^i; \mu_{t+s}^{k,i}, \sigma_{t+s}^{k,i})$$
$$(\hat{\mu}_{t+s}^{k,i}, \hat{\sigma}_{t+s}^{k,i}) = f_{\theta_y}^k(y_{t-w+1:t}^i, w_{t-w+1:t}^i)_s$$

Note that the output of $f_{\theta_y}^k$ is a vector of Gaussian parameters. Thus, for all $s, s' \in \{1, \cdots, h\}$, the Gaussian parameters used to sample $p_{\theta_y}(y_{t+s}^i | x_{t+s}^i = k, y_{t-w+1:t}^i, w_{t-w+1:t}^i)$ and $p_{\theta_y}(y_{t+s'}^i | x_{t+s'}^i = k, y_{t-w+1:t}^i, w_{t-w+1:t}^i)$ are calculated by the same neural-network-based model.

- The prior distribution of the hidden states is provided by a neural network component called $f_{\theta_x}$. Based on $y_{t-w+:t+s-1}^i, w_{-t-w+1:t}^i$, $f_{\theta_x}$ returns the initial distribution and transition matrices of the hidden process:

$$p_{\theta_x}(x_{t+s}^i = k | x_{t+s-1}^i = j, y_{t-w+:t}^i, w_{-t-w+1:t}^i) = \alpha_s^{i,j,k},$$
$$\hat{\alpha}_s^{i,j,k} = f_{\theta_x}(y_{t-w+1:t}^i, w_{t-w+1:t}^i)_{s,j,k}.$$

- Finally, the posterior variational distribution of the hidden states $q_\phi(x_{t+s}^i | y_{t-w+1:t+h}^i, w_{t-w+1:t}^i)$ is learnt by a neural-network-based model named $f_\phi$. Based on $y_{t-w+:t+s-1}^i$ and $w_{-t-w+1:t}^i$, $f_\phi$ returns a matrix with $K \times h$ hidden state probabilities :

$$q_\phi(x_{t+s}^i = k | y_{t-w+:t+h}^i, w_{-t-w+1:t}^i) = \beta_s^{i,k},$$
$$\hat{\beta}_s^{i,k} = f_\phi(y_{t-w+1:t}^i, w_{t-w+1:t}^i)_{k,s}.$$

Architectures used for $(f_{\theta_y}^k)_{1 \le k \le K}, f_{\theta_x}$ and $f_\phi$ can be adjusted depending on the nature of the time series, the forecast horizon, if external signals are available, etc. Architectures used in the experiments section are detailed in Appendix A.2 and Appendix B.2. For completeness, a complete code base gathering the model implementation as well as the training process is publicly provided with this work[1].

## 3 EXPERIMENTS

In this section, we assess the performance of the proposed model on several datasets. The first experiment uses the dataset gathering 10000 fashion time series firstly introduced in David et al. (2022a). In this first experiment, the performance of our algorithm is evaluated and compared with several state-of-the-art methods. Moreover, as external signals are available, we show that the model can correctly leverage them to improve predictions. The proposed method is also evaluated with a collection of 8 reference datasets. This second application shows that the model can be easily applied to a wide variety of forecasting tasks and provide accurate predictions, rivaling state-of-the-art Transformer-based approaches.

### 3.1 FASHION DATASET

#### 3.1.1 FASHION TIME SERIES FORECASTING

A first application of the proposed approach is done on the fashion dataset[2] introduced in David et al. (2022a). This dataset gathers a collection of 10000 weekly time series representing the evolution of the visibility of garments on social media. In addition, each sequence is linked with an external signal representing the visibility of the same garment on a sub sample of influencer users. The intuition is that influencers can adopt fashion items in advance and thus help forecasting methods to better predict the evolution of clothing on mainstream users. This dataset turned out to be well suited to our framework as it shows several specific features.

- The fashion dataset contains numerous time series, showing thousands of different patterns of seasonality, trends, and noise levels. Some of the fashion time series are non-stationary.

---

[1] https://anonymous.4open.science/r/next-302C/
[2] https://github.com/etidav/HERMES/

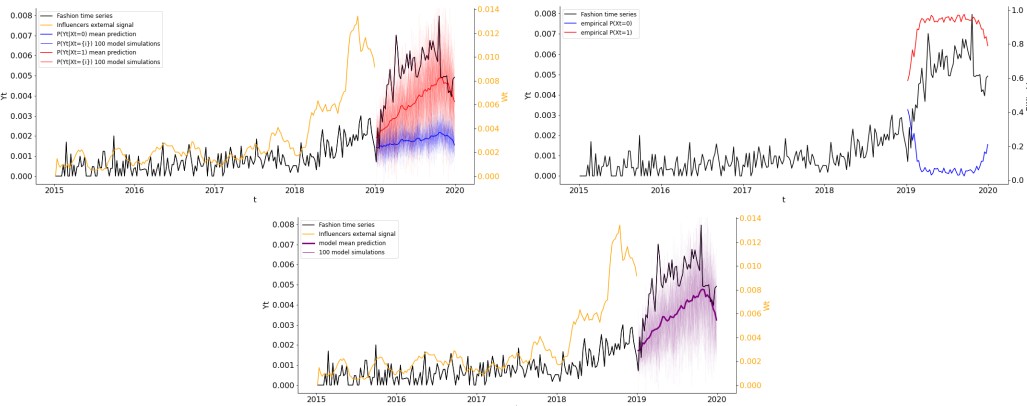

Figure 2: **Predictions on the fashion dataset.** (Top Left) Prediction of the two emission distributions when the hidden state is 0 or 1. (Top Right) Empirical distributions of the hidden states. (Bottom) Simulated predictions with our model using external signals.

- In some examples, early signals announcing the emergence of a new fashion item (which can also be considered as a change of regime) can be perceived in the external signals. Properly exploiting these additional signals could prove decisive in order to accurately detect and predict sudden changes present in the main time series.

### 3.1.2 BASELINE MODELS AND OUR MODEL VARIANTS

The following methods are tested on the fashion dataset as baseline approaches: *Snaive*, *Thetam* (Hyndman et al., 2020), *Ets* (Brown & Meyer, 1961; Holt, 2004), *Tbats* (Livera et al., 2011), *HERMES* (David et al., 2022a), *Prophet* (Taylor & Letham, 2017), *N-BEATS* (Oreshkin et al., 2019), *N-HiTS* (Challu et al., 2023), *DeepAR* (Salinas et al., 2020), *Informer* (Zhou et al., 2021b), *TimesNet* (Wu et al., 2023) and *PathTST* (Nie et al., 2023). All these methods are reviewed in Appendix A.1. Against these methods, 2 variations of our approach with 2 hidden states are presented: i) a variation (mentioned as *Ours*) that does not have access to the influencers external signals. ii) a variation having access to the external signals (mentioned as *Ours-es* with '-es' for external signals). For the second method, external signals are included as input in the hidden state distribution and in only one of the two emission distributions. Further information concerning parameters selection and the training process of the proposed approaches and some of the benchmark methods are reviewed in Appendix A.4.

### 3.1.3 ACCURACY METRICS

The fashion forecasting task is to predict the last year (52 values) of the 10000 time series. Evaluation of the tested methods accuracy is done using the Mean Absolute Scaled Error (MASE) as the fashion time series have different volumes:

$$\text{MASE} = \frac{T - m}{h} \frac{\sum_{j=1}^{h} |Y_{T+j} - \hat{Y}_{T+j}|}{\sum_{i=1}^{T-m} |Y_i - Y_{i-m}|} \, ,$$

where $T$ stands for the time series length, $h$ the forecast horizon and $m$ the seasonality (for the fashion dataset, $T = 209$, $h = 52$ and $m = 52$). In addition to assessing the MASE on the whole dataset, the MASE is also evaluated on 2 sub samples of time series representing stationary and non-stationary time series. The following methodology is used to create these 2 samples.

- **non-stationary time series**. As a main challenge of the fashion forecasting use case is to correctly anticipate sudden evolution, a sub sample of time series showing strong non-stationary behaviours is studied. To create this sub sample of time series, the *snaive* model is used to predict the last year of the fashion time series and the associated MASE are calculated. The non-stationary time series are defined as the 1000 time series where the *snaive* prediction obtained the highest MASE.

Table 1: **Fashion dataset accuracy results.** The Average MASE of each tested method is assessed on the whole dataset and 2 sub samples. For approaches using neural networks, 10 models are trained with different seeds. The mean and standard deviation of the 10 results computed with the 10 replicates are displayed.

| | Fashion dataset | | Non-stationary time series | | Stationary time series | |
|---|---|---|---|---|---|---|
| | *MASE* | *seed std* | *MASE* | *seed std* | *MASE* | *seed std* |
| *Snaive* | 0.881 | - | 1.455 | - | 0.536 | - |
| *Thetam* | 0.844 | - | 1.314 | - | 0.615 | - |
| *Arima* | 0.826 | - | 1.256 | - | 0.565 | - |
| *Ets* | 0.807 | - | 1.270 | - | 0.611 | - |
| *Prophet* | 0.786 | - | 1.193 | - | 0.629 | - |
| *Stlm* | 0.770 | - | 1.198 | - | 0.513 | - |
| *Tbats* | 0.745 | - | 1.229 | - | 0.501 | - |
| *DeepAR* | 0.731 | 0.006 | 1.158 | 0.031 | 0.508 | 0.017 |
| *Informer* | 0.723 | 0.005 | 1.188 | 0.016 | 0.473 | 0.005 |
| *Hermes-ws* | 0.713 | 0.005 | 1.092 | 0.007 | 0.477 | 0.008 |
| *TimesNet* | 0.709 | 0.005 | 1.161 | 0.025 | 0.450 | 0.007 |
| *PatchTST* | 0.706 | 0.004 | 1.149 | 0.001 | 0.448 | 0.003 |
| *N-HiTS* | 0.701 | 0.003 | 1.151 | 0.014 | 0.449 | 0.005 |
| *N-BEATS* | 0.700 | 0.003 | 1.146 | 0.014 | 0.451 | 0.003 |
| *Ours* | 0.692 | 0.001 | 1.116 | 0.006 | **0.440** | 0.001 |
| *Ours-es* | **0.684** | 0.001 | **1.030** | 0.006 | 0.449 | 0.002 |

- **stationary time series**. By contrast, a group of stationary time series is presented. To define them, the same methodology as the previous group is used. We define them as the 1000 time series where the *snaive* prediction reached the lowest MASE.

### 3.1.4 RESULTS

An example of model prediction on a fashion time series is displayed in Figure 2. Hidden states trajectories, emission distributions predictions and the final simulations are presented. In this example, the second emission distribution (that has access to the external signal) catches the regime shift in the time series. This information is also correctly learnt by the hidden states distribution as the empirical probability to be in this regime is close to one. Additional examples are provided in Appendix A.9.

The final accuracy results are provided in Table 1. For each method, a prediction and the associated MASE are computed for the 10000 time series and the average is computed on the whole dataset, the non-stationary sample and the stationary sample. Among methods that do not have access to the external signal, the proposed method (*Ours*) has the highest accuracy on the whole dataset as well as on the 2 sub samples and outperforms other state-of-the-art models. The best results are provided by *Ours-es*, the proposed method with the external signals. It outperforms all the other methods and shows a significant improvement, especially on the non-stationary time series.

### 3.1.5 PROBABILISTIC FORECAST

Compared to many recent Transformer-based methods, the proposed generative model allows sampling trajectories to assess the confidence of the forecast, see Figure 2 for an illustration. So as to evaluate the proposed approach on this specific point, 100 trajectories are computed with the method for each time series. Then, the MASE is computed for each trajectories and the average and standard deviation is displayed in Table 2. As a benchmark, the DeepAR method is used because it also provides probabilistic forecasts allowing sampling of prediction trajectories. We can see that the proposed model outperforms DeepAR and improves its probabilistic predictions when the influencers external signals are used.

Table 2: **Fashion dataset probabilistic forecast accuracy results.** Final accuracy of methods providing probabilistic forecasts. For each method, 100 trajectories and their associated MASEs are computed for each time series. The average and standard deviation is then calculated on the whole dataset and 2 sub samples.

| | Fashion dataset MASE | | Non-stationary time series MASE | | Stationary time series MASE | |
|---|---|---|---|---|---|---|
| | *Mean* | *Std* | *Mean* | *Std* | *Mean* | *Std* |
| *DeepAR* | 0.969 | 0.339 | 1.407 | 0.519 | 0.708 | 0.262 |
| *Ours* | 0.951 | 0.273 | 1.364 | 0.394 | **0.655** | **0.153** |
| *Ours-es* | **0.943** | **0.268** | **1.319** | **0.362** | 0.656 | 0.166 |

## 3.2 REFERENCE DATASET

### 3.2.1 DATASET PRESENTATION, BASELINE MODELS AND THE PROPOSED APPROACH

The presented model is also evaluated with a collection of 8 reference datasets used in many recent contribution dealing with time series forecasting, see Li et al. (2019); Zhou et al. (2021b; 2022); Woo et al. (2022); Wu et al. (2022); Liu et al. (2022); Zeng et al. (2022); Wu et al. (2023); Challu et al. (2023); Woo et al. (2023); Nie et al. (2023). A review of these datasets is given in Appendix B.1.

As benchmark against the proposed models, methods and results presented in the two following recent papers are used Wu et al. (2023); Nie et al. (2023): the two best Transformer-based methods on the 8 datasets named PatchTST (Nie et al., 2023) and TimesNet (Wu et al., 2023), a neural network called Dlinear that, as our approach, only relies and fully connected layers (Zeng et al., 2022) and 5 Transformer-based methods called FEDformer (Zhou et al., 2022), Autoformer (Wu et al., 2022), Informer (Zhou et al., 2021b), Pyraformer (Liu et al., 2022) and LogTrans (Li et al., 2019).

Concerning the proposed approach, recurrent neural networks used in the fashion use case are replaced by fully connected networks as they are too computationally intensive for the long-term forecasting tasks (H=720). For all the reference datasets, the number of hidden states was set to 3 and the same architecture was used. Only a small grid search was run on each dataset for the shortest forecasting task to fix the length of the method inputs. Additional information concerning the proposed model on the reference dataset can be found in Appendix B and a code base is released to reproduce the results[3].

### 3.2.2 ACCURACY METRICS

On the reference datasets, forecasting methods are evaluated on several horizons (lying between 24 to 720 time steps) and with 2 errors metrics, the Mean Square Error (MSE) and the Mean Absolute Error (MAE):

$$\text{MSE} = \frac{1}{h} \sum_{j=1}^{h} (Y_{T+j} - \hat{Y}_{T+j})^2, \qquad \text{MAE} = \frac{1}{h} \sum_{j=1}^{h} |Y_{T+j} - \hat{Y}_{T+j}|,$$

with $h$ the forecast horizon. The last 20% of each time series is kept hidden and used as a test set.

### 3.2.3 RESULTS

Table 3 displays the accuracy results of the benchmark models along with the proposed method on the 8 reference datasets. We can see that depending on the dataset and the horizon, the proposed method and the Transformer-based method PatchTST outperform all alternatives. These results illustrate two important features of the presented approach:

- The proposed method can be used for a large variety of time series forecasting tasks.

---

[3]https://anonymous.4open.science/r/next-302C/

Table 3: **Reference datasets accuracy results.** The best methods are highlighted in bold and the second best results with an underline.

| | H | Ours MSE | Ours MAE | PatchTST/64 MSE | PatchTST/64 MAE | TimesNet MSE | TimesNet MAE | DLinear MSE | DLinear MAE | FEDformer MSE | FEDformer MAE | Autoformer MSE | Autoformer MAE | Informer MSE | Informer MAE | Pyraformer MSE | Pyraformer MAE | LogTrans MSE | LogTrans MAE |
|---|---|---|---|---|---|---|---|---|---|---|---|---|---|---|---|---|---|---|---|
| *Weather* | 96 | 0.154 | 0.199 | **0.149** | **0.198** | 0.172 | 0.220 | 0.176 | 0.237 | 0.217 | 0.296 | 0.266 | 0.336 | 0.300 | 0.384 | 0.896 | 0.556 | 0.458 | 0.490 |
| | 192 | 0.198 | 0.242 | **0.194** | **0.241** | 0.219 | 0.261 | 0.220 | 0.282 | 0.276 | 0.336 | 0.307 | 0.367 | 0.598 | 0.544 | 0.622 | 0.624 | 0.658 | 0.589 |
| | 336 | 0.252 | 0.286 | **0.245** | **0.282** | 0.280 | 0.306 | 0.265 | 0.319 | 0.339 | 0.38 | 0.359 | 0.395 | 0.578 | 0.523 | 0.739 | 0.753 | 0.797 | 0.652 |
| | 720 | 0.316 | **0.332** | 0.314 | 0.334 | 0.365 | 0.359 | 0.323 | 0.362 | 0.403 | 0.428 | 0.419 | 0.428 | 1.059 | 0.741 | 1.004 | 0.934 | 0.869 | 0.675 |
| *Traffic* | 96 | 0.396 | 0.282 | **0.360** | **0.249** | 0.593 | 0.321 | 0.410 | 0.282 | 0.562 | 0.349 | 0.613 | 0.388 | 0.719 | 0.391 | 2.085 | 0.468 | 0.684 | 0.384 |
| | 192 | 0.423 | 0.301 | **0.379** | **0.256** | 0.617 | 0.336 | 0.423 | 0.287 | 0.562 | 0.346 | 0.616 | 0.382 | 0.696 | 0.379 | 0.867 | 0.467 | 0.685 | 0.390 |
| | 336 | 0.437 | 0.306 | **0.392** | **0.264** | 0.629 | 0.336 | 0.436 | 0.296 | 0.570 | 0.323 | 0.622 | 0.337 | 0.777 | 0.420 | 0.869 | 0.469 | 0.733 | 0.408 |
| | 720 | 0.480 | 0.328 | **0.432** | **0.286** | 0.640 | 0.350 | 0.466 | 0.315 | 0.596 | 0.368 | 0.660 | 0.408 | 0.864 | 0.472 | 0.881 | 0.473 | 0.717 | 0.396 |
| *ECL* | 96 | 0.140 | 0.240 | **0.129** | **0.222** | 0.168 | 0.272 | 0.140 | 0.237 | 0.183 | 0.297 | 0.201 | 0.317 | 0.274 | 0.368 | 0.386 | 0.449 | 0.258 | 0.357 |
| | 192 | 0.158 | 0.256 | **0.147** | **0.240** | 0.184 | 0.289 | 0.153 | 0.249 | 0.195 | 0.308 | 0.222 | 0.334 | 0.296 | 0.386 | 0.386 | 0.443 | 0.266 | 0.368 |
| | 336 | 0.176 | 0.274 | **0.163** | **0.259** | 0.198 | 0.300 | 0.169 | 0.267 | 0.212 | 0.313 | 0.231 | 0.338 | 0.300 | 0.394 | 0.378 | 0.443 | 0.280 | 0.380 |
| | 720 | 0.217 | 0.307 | **0.197** | **0.290** | 0.220 | 0.320 | 0.203 | 0.301 | 0.231 | 0.343 | 0.254 | 0.361 | 0.373 | 0.439 | 0.376 | 0.445 | 0.283 | 0.376 |
| *ILI* | 24 | 1.985 | 0.825 | **1.319** | **0.754** | 2.317 | 0.934 | 2.215 | 1.081 | 2.203 | 0.963 | 3.483 | 1.287 | 5.764 | 1.677 | 1.420 | 2.012 | 4.480 | 1.444 |
| | 36 | 1.746 | **0.783** | 1.579 | 0.870 | 1.972 | 0.920 | 1.963 | 0.963 | 2.272 | 0.976 | 3.103 | 1.148 | 4.755 | 1.467 | 7.394 | 2.031 | 4.799 | 1.467 |
| | 48 | 1.722 | 0.790 | **1.553** | 0.815 | 2.238 | 0.940 | 2.130 | 1.024 | 2.269 | 0.981 | 2.669 | 1.085 | 4.763 | 1.469 | 7.551 | 2.057 | 4.800 | 1.468 |
| | 60 | 1.684 | 0.792 | **1.470** | 0.788 | 2.027 | 0.928 | 2.368 | 1.096 | 2.545 | 1.061 | 2.770 | 1.125 | 5.264 | 1.564 | 7.662 | 2.100 | 5.278 | 1.560 |
| *ETTh1* | 96 | 0.379 | **0.389** | 0.370 | 0.400 | 0.384 | 0.402 | 0.375 | 0.399 | 0.376 | 0.419 | 0.449 | 0.459 | 0.865 | 0.713 | 0.664 | 0.612 | 0.878 | 0.740 |
| | 192 | 0.440 | 0.424 | 0.413 | 0.429 | 0.436 | 0.429 | **0.405** | **0.416** | 0.420 | 0.448 | 0.500 | 0.482 | 1.008 | 0.792 | 0.790 | 0.681 | 1.037 | 0.824 |
| | 336 | 0.483 | 0.445 | **0.422** | **0.440** | 0.491 | 0.469 | 0.439 | 0.443 | 0.459 | 0.465 | 0.521 | 0.496 | 1.107 | 0.809 | 0.891 | 0.738 | 1.238 | 0.932 |
| | 720 | 0.570 | 0.524 | **0.447** | **0.468** | 0.521 | 0.500 | 0.472 | 0.490 | 0.506 | 0.507 | 0.514 | 0.512 | 1.181 | 0.865 | 0.963 | 0.782 | 1.135 | 0.852 |
| *ETTh2* | 96 | **0.271** | **0.332** | 0.274 | 0.337 | 0.340 | 0.374 | 0.289 | 0.353 | 0.346 | 0.388 | 0.358 | 0.397 | 3.755 | 1.525 | 0.645 | 0.597 | 2.116 | 1.197 |
| | 192 | 0.347 | **0.382** | **0.341** | 0.382 | 0.402 | 0.414 | 0.383 | 0.418 | 0.429 | 0.439 | 0.456 | 0.452 | 5.602 | 1.931 | 0.788 | 0.683 | 4.315 | 1.635 |
| | 336 | 0.380 | 0.409 | **0.329** | **0.384** | 0.452 | 0.452 | 0.448 | 0.465 | 0.496 | 0.487 | 0.482 | 0.486 | 4.721 | 1.835 | 0.907 | 0.747 | 1.124 | 1.604 |
| | 720 | 0.420 | 0.446 | **0.379** | **0.422** | 0.462 | 0.468 | 0.605 | 0.551 | 0.463 | 0.474 | 0.515 | 0.511 | 3.647 | 1.625 | 0.963 | 0.783 | 3.188 | 1.540 |
| *ETTm1* | 96 | **0.288** | **0.335** | 0.293 | 0.346 | 0.338 | 0.375 | 0.299 | 0.343 | 0.379 | 0.419 | 0.505 | 0.475 | 0.672 | 0.571 | 0.543 | 0.510 | 0.600 | 0.546 |
| | 192 | **0.331** | **0.363** | 0.333 | 0.370 | 0.374 | 0.387 | 0.335 | 0.365 | 0.426 | 0.441 | 0.553 | 0.496 | 0.795 | 0.669 | 0.557 | 0.537 | 0.837 | 0.700 |
| | 336 | **0.364** | **0.385** | 0.369 | 0.392 | 0.410 | 0.411 | 0.369 | 0.386 | 0.445 | 0.459 | 0.621 | 0.537 | 1.212 | 0.871 | 0.754 | 0.655 | 1.124 | 0.832 |
| | 720 | 0.429 | 0.426 | **0.416** | **0.420** | 0.478 | 0.450 | 0.425 | 0.421 | 0.543 | 0.490 | 0.671 | 0.561 | 1.166 | 0.823 | 0.908 | 0.724 | 1.153 | 0.820 |
| *ETTm2* | 96 | **0.162** | **0.249** | 0.166 | 0.256 | 0.187 | 0.267 | 0.167 | 0.260 | 0.203 | 0.287 | 0.255 | 0.339 | 0.365 | 0.453 | 0.435 | 0.507 | 0.768 | 0.642 |
| | 192 | **0.218** | **0.288** | 0.223 | 0.296 | 0.249 | 0.309 | 0.224 | 0.303 | 0.269 | 0.328 | 0.281 | 0.340 | 0.533 | 0.563 | 0.730 | 0.673 | 0.989 | 0.757 |
| | 336 | **0.271** | **0.325** | 0.274 | 0.329 | 0.321 | 0.351 | 0.281 | 0.342 | 0.325 | 0.366 | 0.339 | 0.372 | 1.363 | 0.887 | 1.201 | 0.845 | 1.334 | 0.872 |
| | 720 | **0.355** | **0.380** | 0.362 | 0.385 | 0.408 | 0.403 | 0.397 | 0.421 | 0.421 | 0.415 | 0.422 | 0.419 | 3.379 | 1.388 | 3.625 | 1.451 | 3.048 | 1.328 |

- By combining elementary neural networks components with hidden processes and a computationally efficient training procedure, our approach reaches state-of-the-art standards while providing uncertainty quantification.

However, we can see that the Transformer-based model PatchTST outperforms the proposed model on some reference datasets such as Traffic or ECL. A main reason is that these two datasets gather similar time series with long term evolution. In this context, the interest in introducing hidden states is low and the models used in the emission distributions do not manage to outperform complex and high dimensional models such as PatchTST. Future work could focus on understanding how to overcome this issue, by refining the training process and/or introducing Transformer-based architectures into the proposed model and variational approximation. Additional numerical results on the reference datasets can be found in Appendix B along with examples of predictions for each reference dataset.

## 4  CONCLUSION

In this paper, we introduced a new time series forecasting model combining discrete hidden Markov models and deep architectures. We proposed a two-stage training procedure, based on the ELBO and inspired by recent variational quantization approaches. Our model outperformed state-of-the-art methods in particular when using external signals on fashion time series. Then, its performance were assessed on 8 reference datasets with similar performance as other state-of-the-art methods.

The numerical performance of our method makes it a solid alternative to recent Tansformer-based models. Furthermore, unlike many other state-of-the-art methods, it allows an estimation of the predictive distribution of future observations. Finally, we want to highlight that several parts of our approach can be further investigated to improve the performance presented such as providing an automatic selection of the number of hidden states, or extending recent results on variational learning of hidden Markov models to obtain theoretical guarantees on the variational distribution.

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

## A FASHION DATASET

### A.1 BENCHMARK MODELS

We present in this section the baseline methods tested on the fashion dataset against the proposed approach:

- **Snaive:** A method that only repeats the last past period of historical data.
- **Thetam:** A parametric model that decomposes the original signal in $\theta$-lines, predicts each one separately and recomposes them to produce the final forecast (Hyndman et al., 2020).
- **Ets:** The exponential smoothing method (Brown & Meyer, 1961; Holt, 2004).
- **Tbats:** A parametric model presented in (Livera et al., 2011).
- **Stlm:** A parametric model that uses a multiplicative decomposition and models the seasonally adjusted time series with an exponential smoothing model (Hyndman et al., 2020).
- **HERMES:** a hybrid method mixing per-time-series TBATS predictors and a recurrent neural network global corrector (David et al., 2022a).
- **Prophet:** a parametric model introduced in Taylor & Letham (2017) and widely used in the industry.
- **N-BEATS:** a full-neural-network-based method that shows striking results on numerous datasets of the literature (Oreshkin et al., 2019).
- **N-HiTS:** The evolution of N-BEATS (Challu et al., 2023).
- **DeepAR:** a full-neural-network-based method used at Amazon that provided sharp probabilistic forecasts (Salinas et al., 2020).
- **Informer:** A Transformer-based model proposing a new self-attention mechanism reducing the Transformers' high memory usage (Zhou et al., 2021b).
- **TimesNet:** One of the most recent Transformer-based models (Wu et al., 2023).

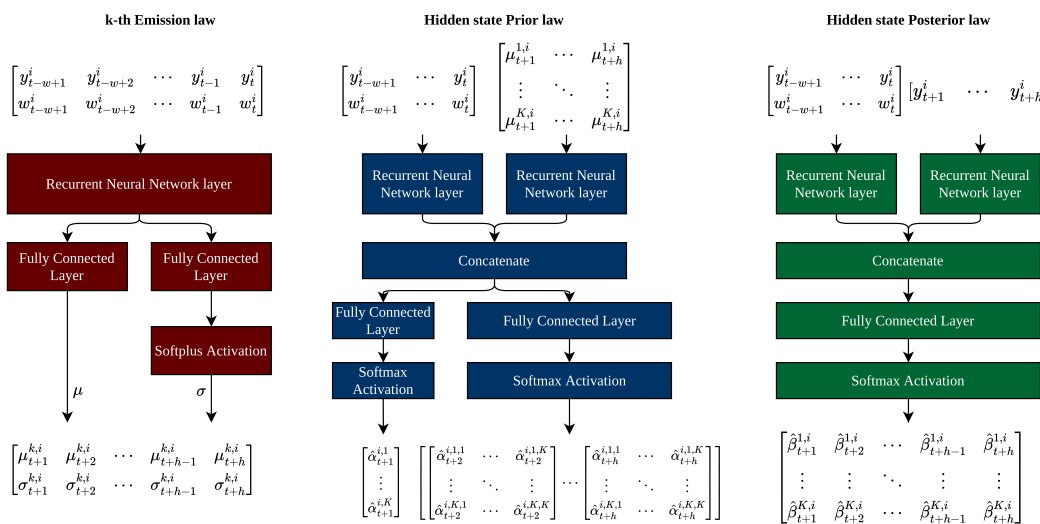

Figure 3: **Example of model architecture.** Example of architecture used for the proposed approach on the Fashion dataset. (Left) Model used to compute parameters of the k-th emission distribution. (Middle) Model used to compute the hidden state probabilities. (Right) Model used to approximate the posterior distribution of the hidden states.

- **PatchTST:** A Transformer-based model that emerged as the best method using Transformers on several datasets of the literature (Nie et al., 2023).

For the methods DeepAR, N-BEATS, N-HiTS, Informer, TimesNet and PatchTST, the package "neuralforecast" was used to train them (Olivares et al., 2022).

## A.2 ARCHITECTURE USED FOR HIDDEN STATES AND EMISSION DISTRIBUTIONS

We detail in this section the architecture used for the proposed model on the Fashion dataset. For the emission distributions, a LSTM layer is first used to process the past of the main signal and external signals. Then, Fully Connected (FC) layers are used to compute the different parameters of the emission distributions. For the standard deviation of the Gaussian emission distributions, a "softplus" activation is applied on the last layer to ensure that the model outputs remain positive. Concerning the hidden state prior distribution, two LSTM layers are used to process the past inputs (main signals plus external signals) and some outputs of the emission distributions. Outputs are concatenated and fed to two Fully Connected layers followed by a "softmax" activation to compute the initial distribution and transition matrices of the hidden state processes. Finally, for the posterior distribution of the hidden states, past and future windows of the main signal are first provided to two LSTM layers. Outputs are concatenated and fed to a Fully Connected layer and a "softmax" activation to compute the posterior variational probabilities. See Figure 3 for an illustration of the different components.

## A.3 FIXING THE NUMBER OF HIDDEN STATES

Table 4 displays results of several variations of the proposed model with a number of hidden states between 2 and 4. We see that the variation achieving the best accuracy is the method with two hidden states and that increasing the number of hidden states does not always lead to an increase in final accuracy. Indeed, we notice that the more the model has hidden states, the more difficult it is to differentiate them, which leads to redundant emission distributions. See Appendix A.9 for prediction examples of the variation with 4 hidden states. Future works will focus on providing an automatic selection of $K$.

Table 4: **Hidden states grid search.** Average MASE of the proposed model with a number of hidden states lying between 2 and 4 are assessed on the Fashion dataset. For each model, 10 models are trained with different seeds. The mean and standard deviation of the 10 results computed with the 10 replicates are displayed.

| | Fashion dataset | | Non-stationary time series | | Stationary time series | |
|---|---|---|---|---|---|---|
| | *MASE* | *seed std* | *MASE* | *seed std* | *MASE* | *seed std* |
| *Ours 3hs* | 0.693 | 0.001 | 1.118 | 0.006 | 0.441 | 0.002 |
| *Ours 4hs* | 0.693 | 0.001 | 1.113 | 0.004 | 0.442 | 0.001 |
| *Ours 2hs* | 0.692 | 0.001 | 1.116 | 0.006 | **0.44** | 0.001 |
| *Ours-es 3hs* | 0.685 | 0.001 | 1.031 | 0.005 | 0.452 | 0.002 |
| *Ours-es 4hs* | 0.685 | 0.001 | **1.029** | 0.005 | 0.452 | 0.002 |
| *Ours-es 2hs* | **0.684** | 0.001 | 1.03 | 0.006 | 0.449 | 0.002 |

Table 5: **Fashion dataset benchmarks grid search** Grid searches run on the Fashion dataset for the following benchmark methods: DeepAR, PatchTST, N-HiTS, N-BEATS and our method. The metrics displayed are the final MASE of each model variation on the test set.

*DeepAR*

| | | Learning rate | | |
|---|---|---|---|---|
| | | *0.005* | *0.0005* | *0.00005* |
| Batch size | 8 | 0.76 | 0.791 | 0.874 |
| | 64 | 0.733 | 0.736 | 0.831 |
| | 256 | 0.774 | 0.771 | 0.772 |
| | 1024 | 0.754 | 0.75 | 0.754 |
| | 2048 | **0.727** | 0.745 | 0.752 |

*PatchTST*

| | | Learning rate | | |
|---|---|---|---|---|
| | | *0.005* | *0.0005* | *0.00005* |
| Batch size | 8 | 0.85 | 0.714 | 0.717 |
| | 64 | 0.881 | 0.705 | 0.707 |
| | 256 | 0.913 | 0.705 | 0.708 |
| | 1024 | 0.818 | **0.704** | 0.709 |
| | 2048 | 0.947 | 0.709 | 0.709 |

*Ours*

| | | Learning rate | | |
|---|---|---|---|---|
| | | *0.005* | *0.0005* | *0.00005* |
| Batch size | 64 | 0.714 | 0.713 | 0.727 |
| | 256 | 0.703 | 0.700 | 0.714 |
| | 1024 | 0.694 | 0.696 | 0.709 |
| | 2048 | 0.694 | **0.693** | 0.702 |

*N-HiTS*

| | | Learning rate | | |
|---|---|---|---|---|
| | | *0.005* | *0.0005* | *0.00005* |
| Batch size | 8 | 0.733 | 0.709 | **0.701** |
| | 64 | 0.716 | 0.733 | 0.702 |
| | 256 | 0.719 | 0.733 | 0.702 |
| | 1024 | 0.715 | 0.734 | 0.702 |
| | 2048 | 0.717 | 0.734 | 0.703 |

*N-BEATS*

| | | Learning rate | | |
|---|---|---|---|---|
| | | *0.005* | *0.0005* | *0.00005* |
| Batch size | 8 | 0.740 | 0.709 | **0.700** |
| | 64 | 0.713 | 0.734 | 0.702 |
| | 256 | 0.719 | 0.738 | 0.703 |
| | 1024 | 0.718 | 0.737 | 0.704 |
| | 2048 | 0.876 | 0.741 | 0.704 |

*Informer*

| | | Learning rate | | |
|---|---|---|---|---|
| | | *0.005* | *0.0005* | *0.00005* |
| Batch size | 8 | 0.786 | 0.747 | 0.752 |
| | 64 | 0.752 | 0.727 | 0.734 |
| | 256 | 0.736 | 0.719 | 0.735 |
| | 1024 | 0.746 | 0.722 | 0.727 |
| | 2048 | 0.735 | **0.719** | 0.727 |

## A.4 GRID SEARCH

So as to produce the final results of the benchmark methods and the proposed model on the Fashion dataset, several grid searches were run to fix the different hyper parameters. For the methods PatchTST, Informer, N-HiTS, N-BEATS, DeepAR and our method, a grid search was run on the learning rate and the batch size. Table 5 summarizes the grid search results for these 5 models. The best configuration in terms of MASE on the test set was selected and used to produce the final results displayed in Section 3.1.

## A.5 IMPACT OF MODEL SIZE

In addition to the learning rate and the batch size, we also tested different sizes for our approach on the Fashion dataset. Figure 4 displays the evolution of the MASE of our model depending on its size (number of parameters, from 150 thousand parameters to 1.5 millions). We can see that the best MASE is reached by a model with 700 thousand parameters: $2 \times 110000$ parameters for the two emission distributions, 160000 parameters for the hidden states posterior model and 320000 parameters for the hidden states prior model. For comparison, Table 6 displays the size of several other methods used on the fashion dataset (architecture where the horizon is set to 52 and the input size is set to 104.)

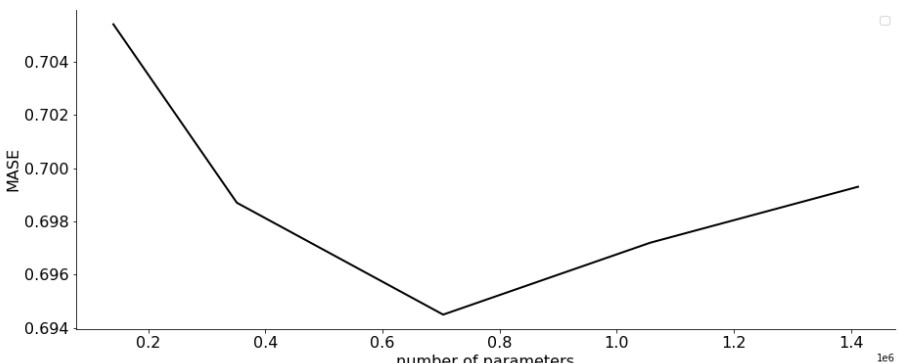

Figure 4: **Accuracy depending on the model size** Evolution of the accuracy of the proposed model on the Fashion dataset depending on its size.

Table 6: **Models size on the Fashion dataset:** A comparison of the size of different methods on the mode dataset in the configuration where the horizon is set to 52 and the input size is set to 104.

|  | Model size |
|---|---|
|  | *number of parameters* |
| *DeepAR* | 199565 |
| *Informer* | 341985 |
| *PatchTST* | 487863 |
| *Ours* | 703598 |
| *Ours-es* | 703598 |
| *N-HiTS* | 2678675 |
| *N-BEATS* | 2729322 |
| *TimesNet* | 4704125 |

A.6    JOINT TRAINING INSTABILITIES

In this section, we investigate the impact of the two-steps training process proposed with our approach. This process inspired by ideas brought with Vector quantized VAE model (van den Oord et al., 2018) reveals to be central in the case of time series forecasting. To illustrate this point, we tested variants of our model on the Fashion dataset where all the components were jointly trained. Table 7 presents results of these variants (called *Ours (one-step training)* and *Ours-es (one-step training)*) and highlights that these variants underperform our approaches optimized with a two-step process. In this use case, the hidden state posterior and prior distributions converged to the same deteriorated deterministic distributions, i.e hidden states trajectories are all the same, for every time series. Consequently, the emission distributions are always activated to predict the same parts of the forecasting horizon, for every time series and can not be specialized at forecasting specific behaviours. In addition of Table 7, the ELBO was also evaluated on the eval set and compared between the two training processes. We found that the final ELBO on the evaluation set was always better at the end of the two-stage training than that obtained with the single-stage training. Finally, we tested to repeat several times the two-steps training to improve even more the ELBO. We found that repeating the two-steps training did not lead to an improvement in terms of ELBO and MASE and has an important cost in terms of computational time.

Table 7: **One-step versus two-steps training.** Evaluation of the proposed method on the fashion dataset where a one-step training process versus a two-steps training is used. The Average MASE of each tested method is assessed on the whole dataset and 2 sub samples. For all the variants, 10 models are trained with different seeds. The mean and standard deviation of the 10 results computed with the 10 replicates are displayed.

| | Fashion dataset | | Non-stationary time series | | Stationary time series | |
|---|---|---|---|---|---|---|
| | *MASE* | *seed std* | *MASE* | *seed std* | *MASE* | *seed std* |
| *Ours (one-step-training)* | 0.701 | 0.001 | 1.13 | 0.005 | 0.447 | 0.002 |
| *Ours-es (one-step-training)* | 0.698 | 0.003 | 1.084 | 0.017 | 0.456 | 0.002 |
| *Ours* | 0.692 | 0.001 | 1.116 | 0.006 | **0.440** | 0.001 |
| *Ours-es* | **0.684** | 0.001 | **1.030** | 0.006 | 0.449 | 0.002 |

Table 8: **Leant versus fixed standard deviation.** Evaluation of the proposed method on the fashion dataset where the standard deviation of the emission distributions is not learnt and fixed to 1. The Average MASE of each tested method is assessed on the whole dataset and 2 sub samples. For all the variants, 10 models are trained with different seeds. The mean and standard deviation of the 10 results computed with the 10 replicates are displayed.

| | Fashion dataset | | Non-stationary time series | | Stationary time series | |
|---|---|---|---|---|---|---|
| | *MASE* | *seed std* | *MASE* | *seed std* | *MASE* | *seed std* |
| *Ours (fix std)* | 0.716 | 0.003 | 1.141 | 0.006 | 0.462 | 0.005 |
| *Ours-es (fix std)* | 0.710 | 0.002 | 1.084 | 0.012 | 0.468 | 0.003 |
| *Ours* | 0.692 | 0.001 | 1.116 | 0.006 | **0.440** | 0.001 |
| *Ours-es* | **0.684** | 0.001 | **1.030** | 0.006 | 0.449 | 0.002 |

## A.7 LEARN OR FIX THE EMISSION DISTRIBUTIONS VARIANCES

Compared to several recent forecasting methods, the proposed approach learns emission distributions allowing an estimation of the predictive distribution of the future observations. In this section, we evaluate if learning the emission distributions (mean and standard deviation in the Gaussian case) can have an impact on the final accuracy of the model. To do so, we trained a variant of the proposed approach where we fixed the standard deviation of the emission distributions to 1 and only learned the mean parameters. Table 8 displays results of these alternatives (called *Ours (fix std)* and *Ours-es (fix std)*) alongside the models where the standard deviations are learnt. We can see that freezing the standard deviations has a negative impact on the final accuracy of our approach and it reduces only marginally the model complexity and the learning process time.

## A.8 THE IMPORTANCE OF TEMPORAL HIDDEN STATES

As a last ablation study on the proposed approach, we evaluate the impact of using temporal hidden states, able to switch during the prediction period. To do so, a variant of the proposed approach where the transition matrices of the hidden state are fixed to identity matrices is trained. Thus, for each prediction, the hidden state is only determined by the initial distribution of the hidden states and remains constant during the whole predictive time frame. Table 9 displays results of these alternatives (called *Ours (fix hidden states)* and *Ours-es (fix hidden states)*) alongside our models with moving hidden states. We can see that the Markov dynamics enabling our approach to switch from a hidden state to another during the predictive period have a positive impact on the final accuracy, allowing our approach to capture finer details and better leverage external signals.

Table 9: **Constant versus temporal hidden states.** Evaluation of the proposed method on the fashion dataset where the hidden states are forced to stay constant during the whole forecasting period. The Average MASE of each tested method is assessed on the whole dataset and 2 sub samples. For all the variants, 10 models are trained with different seeds. The mean and standard deviation of the 10 results computed with the 10 replicates are displayed.

| | Fashion dataset | | Non-stationary time series | | Stationary time series | |
|---|---|---|---|---|---|---|
| | *MASE* | *seed std* | *MASE* | *seed std* | *MASE* | *seed std* |
| *Ours (fix hidden states)* | 0.700 | 0.001 | 1.135 | 0.004 | 0.446 | 0.001 |
| *Ours-es (fix hidden states)* | 0.696 | 0.002 | 1.051 | 0.008 | 0.462 | 0.003 |
| *Ours* | 0.692 | 0.001 | 1.116 | 0.006 | **0.440** | 0.001 |
| *Ours-es* | **0.684** | 0.001 | **1.030** | 0.006 | 0.449 | 0.002 |

### A.9 EXAMPLE OF PREDICTIONS

Finally, we display additional examples of predictions of our proposed model on the Fashion dataset. First, Figure 5 displays a comparison between predictions of the model having access to the influencers external signals and without having access to them. We can see that in some examples, the inclusion of the external signals greatly helps one of the emission distributions to explore new distributions and accurately catch non-stationary evolution. Then, Figure 6 shows the prediction of the presented model with 4 hidden states and illustrates that adding hidden states does not necessarily lead to a better exploration of the dataset but may lead to redundant regimes. Finally, Figure 7 displays examples of prediction of the proposed model along with some of the best benchmark models.

## B REFERENCE DATASET

### B.1 REFERENCE DATASETS

We present in this section the 8 references dataset used in Section 3.2.

- **ETTm2** (Electricity Transformer Temperature): a dataset gathering time series following characteristics of an electricity transformer in China from July 2016 to July 2018 with values measured every 15 minutes Zhou et al. (2021a).

- **ECL**(Electricity): time series representing the evolution of the electricity consumption of 370 clients from 2012 to 2014 Trindade (2015).

- **Exchange-Rate**: a dataset gathering 8 time series representing the evolution from 1990 to 2016 of the daily exchange rates of the following countries: Australia, British, Canada, Switzerland, China, Japan, New Zealand and Singapore Lai et al. (2018).

- **Traffic** (San Francisco Bay Area Highway Traffic): 862 time series representing road occupancy measured by 862 sensors spread over the State of California from January 2015 to December 2016.

- **Weather**: dataset gathering the evolution of 21 meteorological variables in Germany during the year 2020.

- **ILI**(Influenza-like illness): time series representing the weekly evolution of the number of influenza-like illness patients in The United States, from January 2002 to July 2020.

### B.2 ARCHITECTURE USED FOR HIDDEN STATES AND EMISSION DISTRIBUTIONS

An overview of the architecture used for the proposed model on the 8 reference dataset is displayed in Figure 8. As recurrent neural layers considerably slow down the model for some of the long-term forecasting tasks (especially where horizon=720), they are all replaced by fully connected layers. Except for this modification, the architecture used with the reference datasets is similar to that used on the Fashion dataset.

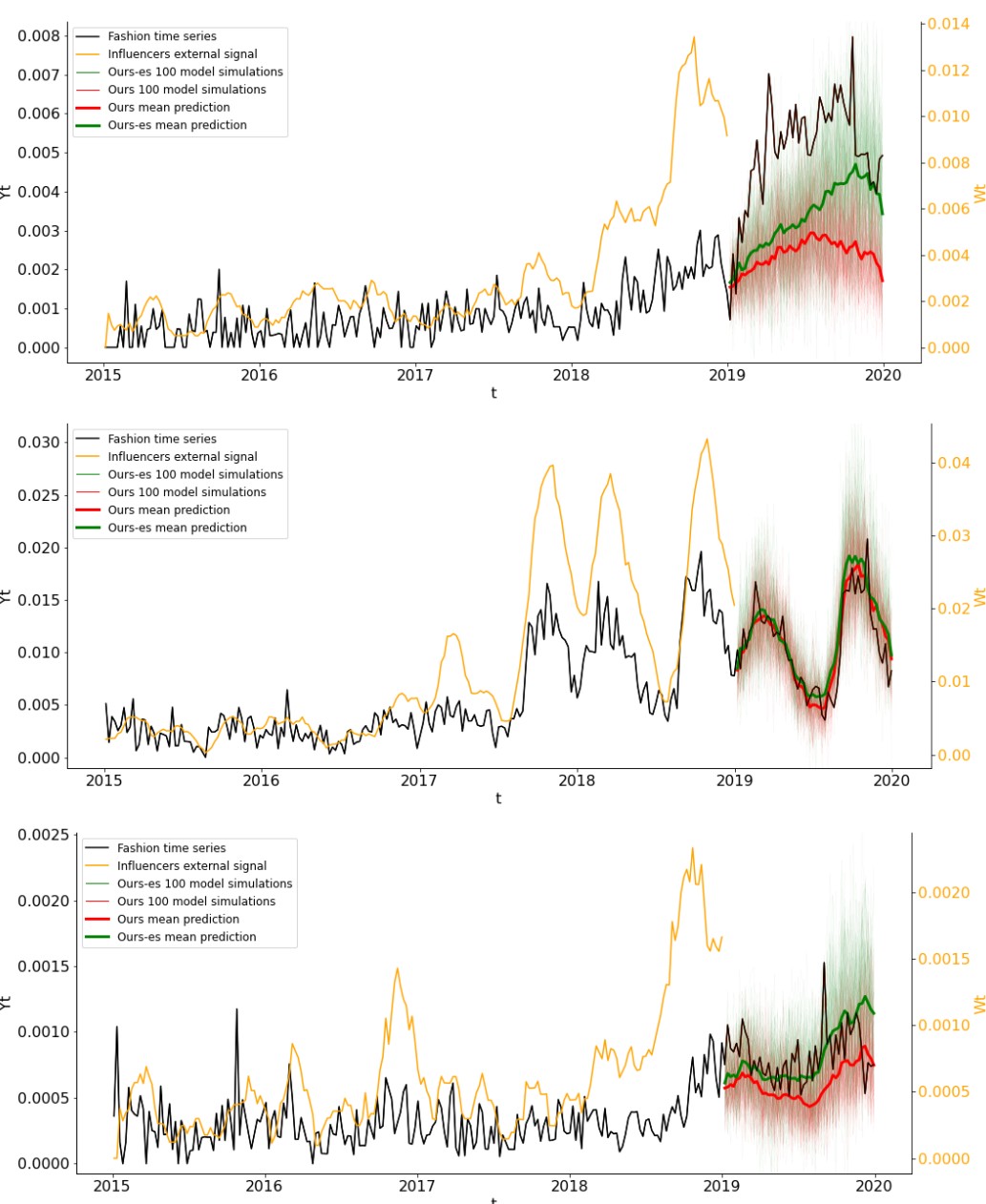

Figure 5: **Ours vs Ours-es predictions.** *Ours* and *Ours-es* model predictions on three fashion time series: (Top) "br_female_shoes_262", (Middle) "eu_female_outerwear_177", (Bottom) "eu_female_texture_80". On several fashion time series, *Ours-es* correctly leverages the influencers external signal and capture sudden non-stationary evolution impossible to forecast without them.

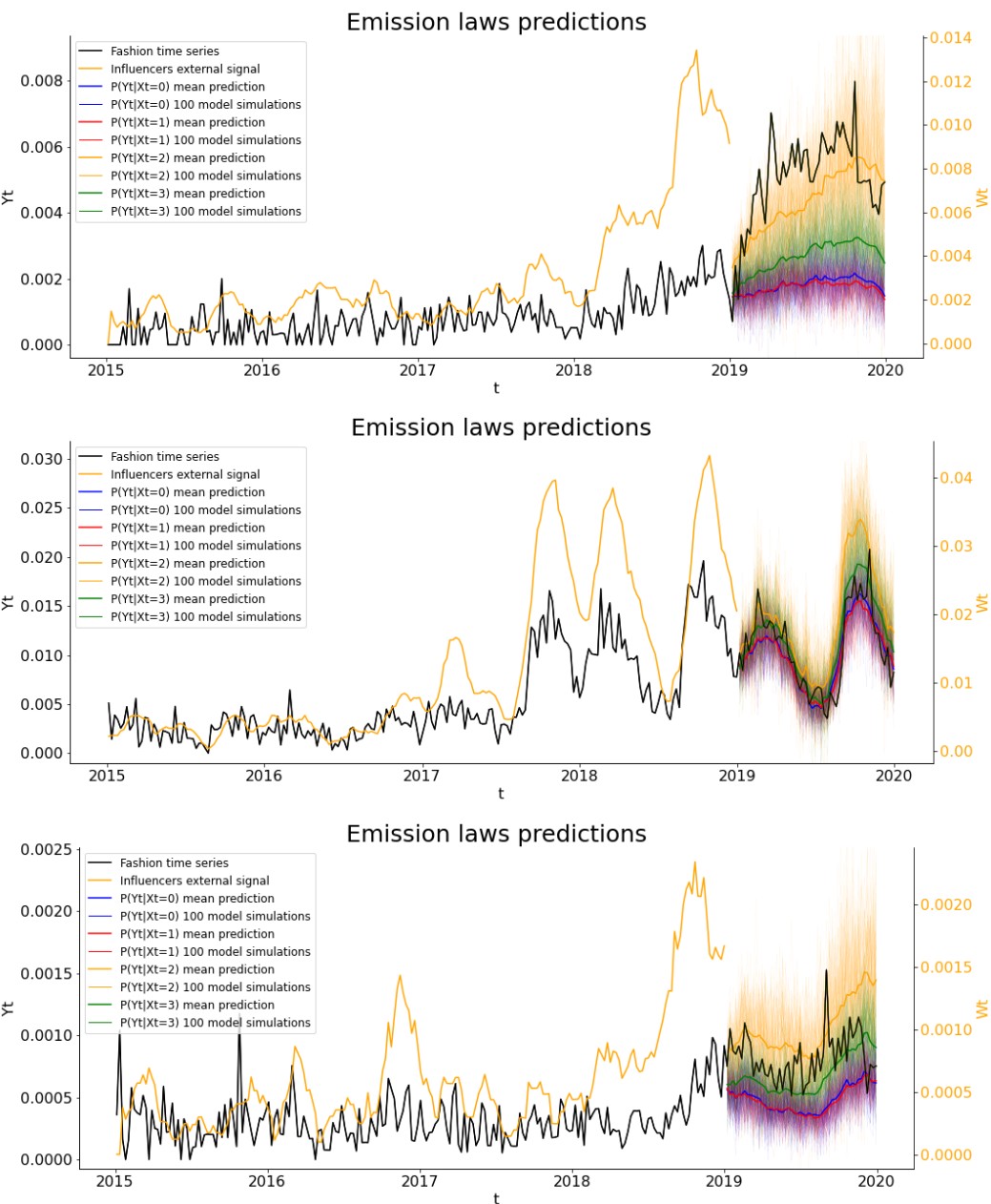

Figure 6: **Proposed method with 4 hidden states predictions.** Emission distributions predictions of the proposed model with 4 hidden states on three fashion time series: (Top) "br_female_shoes_262", (Middle) "eu_female_outerwear_177", (Bottom) "eu_female_texture_80". For this model, the influencers external signal was only given to the third and the fourth emission distributions. The third and the fourth emission distributions learned different distributions but the first and the second ones seem to be redundant.

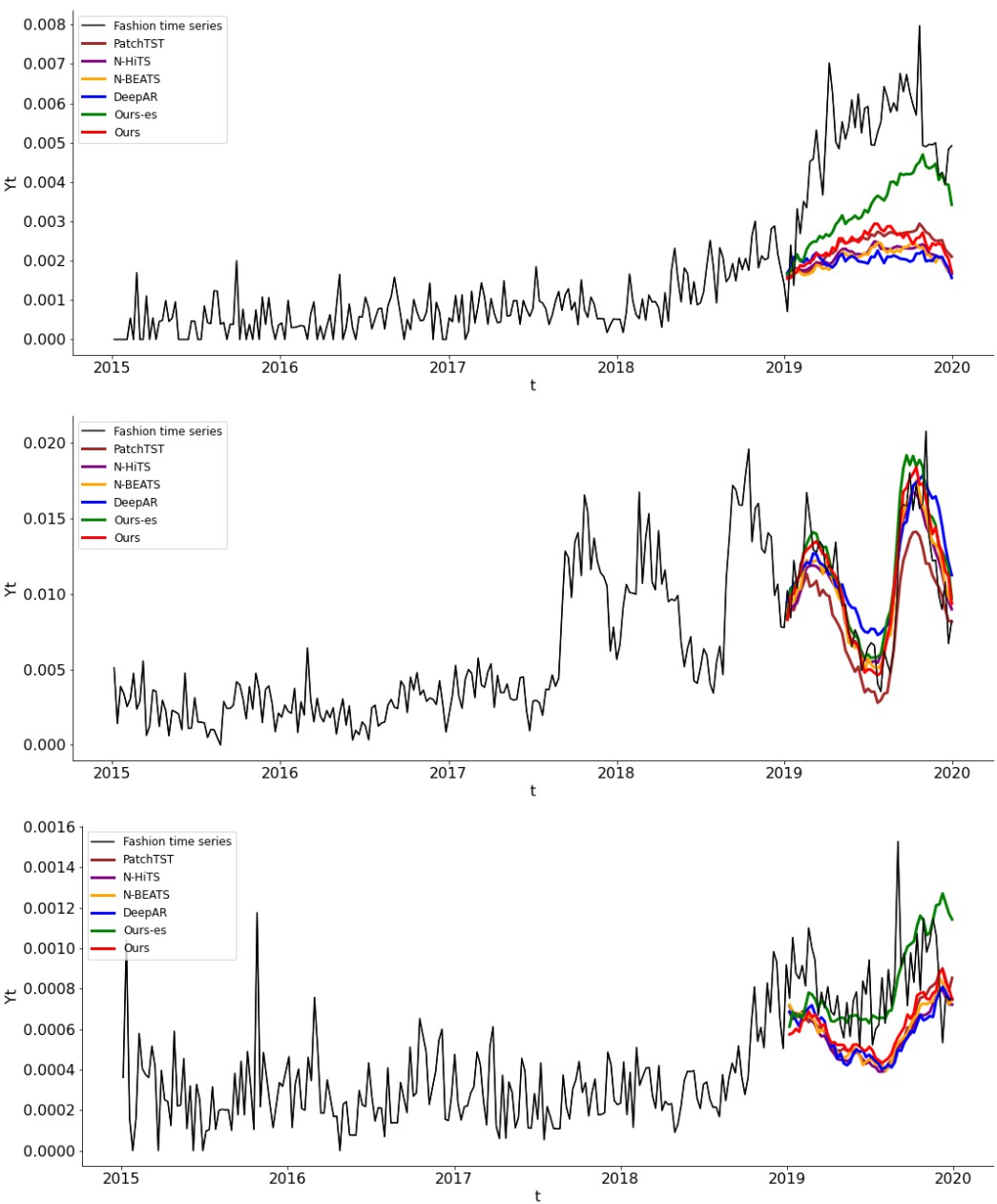

Figure 7: **Presented method and benchmark models predictions.** Final prediction of the presented model and some of the benchmark methods on three fashion time series (Top) "br_female_shoes_262", (Middle) "eu_female_outerwear_177", (Bottom) "eu_female_texture_80". The model *Ours* seems to compute more accurate predictions than benchmark methods on these examples but the best forecasts are provided by the model *Ours-es* with the use of the influencers external signal.

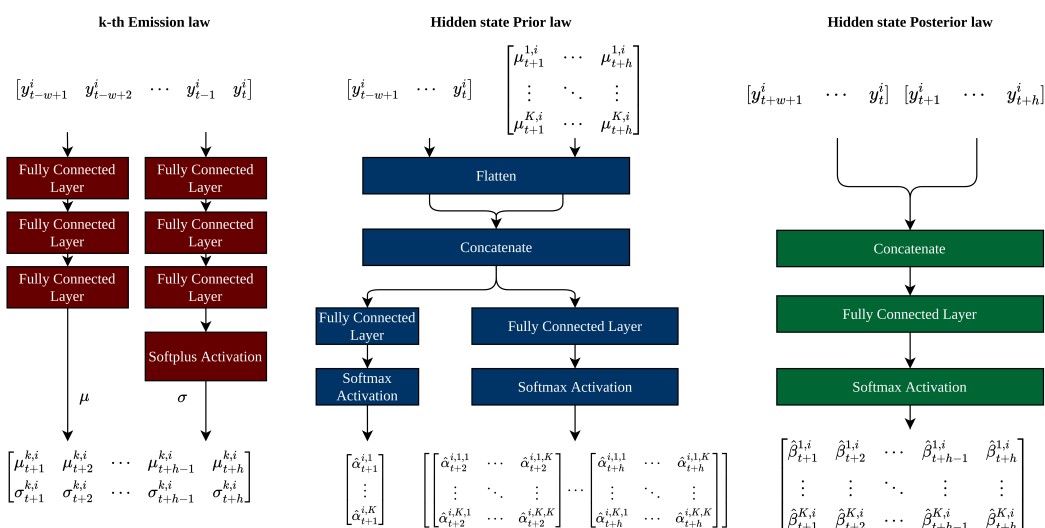

Figure 8: **Example of model architecture.** Example of architecture used for the proposed approach on the 8 reference datasets. (Left) Model used to compute parameters of the k-th emission distribution. (Middle) Model used to compute the hidden state probabilities. (Right) Model used to approximate the posterior variational distribution of the hidden states.

Table 10: **Hidden state parameter** Analysis of the importance of the number of hidden states on 3 of the 8 reference datasets. We test a number of hidden states from 1 to 4. The metrics displayed are the final MSE and MAE on the validation and test set.

| hidden states | | 1 hidden state | | | | 2 hidden states | | | | 3 hidden states | | | | 4 hidden states | | | |
|---|---|---|---|---|---|---|---|---|---|---|---|---|---|---|---|---|---|
| | | Eval | | Test | | Eval | | Test | | Eval | | Test | | Eval | | Test | |
| dataset | H | MSE | MAE | MSE | MAE | MSE | MAE | MSE | MAE | MSE | MAE | MSE | MAE | MSE | MAE | MSE | MAE |
| Weather | 96 | 0.399 | 0.280 | 0.155 | 0.201 | 0.397 | 0.278 | 0.154 | 0.200 | 0.398 | 0.276 | **0.153** | **0.199** | 0.399 | 0.277 | 0.154 | 0.200 |
| Traffic | 96 | 0.329 | 0.244 | **0.399** | 0.287 | 0.330 | 0.243 | 0.400 | 0.286 | 0.330 | 0.242 | **0.399** | **0.284** | 0.330 | 0.242 | **0.399** | 0.285 |
| ETTh2 | 96 | 0.236 | 0.316 | 0.273 | 0.334 | 0.239 | 0.313 | 0.276 | 0.332 | 0.238 | 0.311 | 0.273 | **0.331** | 0.236 | 0.310 | **0.272** | **0.331** |

## B.3 THE IMPORTANCE OF HIDDEN STATES

On all the 8 benchmark datasets, we fix the number of hidden states to 3 for the proposed model. However, as for the past dependency parameter, a gridsearch can be done to find the optimal number of hidden states. On three reference datasets (Traffic, Weather and ETTh2) and for the forecasting task where horizon is fixed to 96, we train 4 variations of the proposed method with a number of hidden states lying between 1 to 4. Results are displayed in Figure 10. We can see that the optimal number of hidden states can change depending on the use case but the difference in terms of accuracy remains low between 2 and 4 hidden states.

## B.4 MINMAXSCALER VERSUS STANDARDSCALER

On the 8 reference datasets, we investigate the potential impact of the preprocessing step on the proposed model. Consequently, on 4 of the 8 reference datasets (ETTh1, ETTh2, ETTm1 and ETTm2) and for the forecasting task where horizon is fixed to 96, the two normalization included by the proposed model (Minmaxscaler and StandardScaler) are tested. Table 11 displays results of the different trainings and we can see that accuracy results can be strongly impacted by the preprocessing. As the Minmaxscaler normalization seems to be more robust than the Standardscaler, it was selected for the proposed architecture on the 8 reference datasets.

Table 11: **Preprocessing analysis** Analysis of the impact of the preprocessing on the proposed method. The Minmaxscaler (scale the inputs between 0 and 1) and the Standardscaler (scale the inputs to have mean 0 and a variance of 1) approach are tested. The metrics displayed are the final MSE and MAE on the validation and test set.

| *preprocess name* | | MinMaxscaler | | | | Standardscaler | | | |
|---|---|---|---|---|---|---|---|---|---|
| | | Eval | | Test | | Eval | | Test | |
| dataset | H | MSE | MAE | MSE | MAE | MSE | MAE | MSE | MAE |
| ETTh1 | 96 | **0.487** | **0.451** | 0.379 | 0.389 | 0.494 | 0.458 | 0.380 | 0.394 |
| ETTh2 | 96 | **0.239** | **0.315** | 0.271 | 0.332 | 0.271 | 0.348 | 0.306 | 0.362 |
| ETTm1 | 96 | **0.303** | **0.354** | 0.288 | 0.336 | 0.304 | 0.355 | 0.292 | 0.340 |
| ETTm2 | 96 | **0.124** | **0.233** | 0.162 | 0.249 | 0.134 | 0.241 | 0.167 | 0.253 |

Table 12: **Past dependency grid search** Grid searches run on the 8 reference datasets to fix the optimal past dependency parameter for the proposed model. We tested a range of values between the half of the seasonality to 8 times the seasonality. The metrics displayed are the final MSE and MAE on the validation set.

| *Past dependency* | | 0.5*seasonality | | 1*seasonality | | 2*seasonality | | 3*seasonality | | 4*seasonality | | 5*seasonality | |
|---|---|---|---|---|---|---|---|---|---|---|---|---|---|
| dataset | H | MSE | MAE | MSE | MAE | MSE | MAE | MSE | MAE | MSE | MAE | MSE | MAE |
| Weather | 96 | 0.639 | 0.346 | 0.470 | 0.290 | 0.433 | 0.277 | 0.423 | **0.276** | 0.409 | **0.276** | **0.400** | **0.276** |
| Traffic | 96 | 0.537 | 0.333 | 0.366 | 0.254 | 0.337 | 0.243 | 0.329 | 0.241 | 0.328 | 0.240 | **0.324** | **0.239** |
| ECL | 96 | 0.183 | 0.256 | 0.132 | 0.225 | 0.122 | 0.217 | 0.119 | **0.215** | **0.118** | **0.215** | **0.118** | 0.216 |
| ILI | 96 | 0.322 | 0.405 | **0.146** | **0.232** | 0.296 | 0.317 | 0.237 | 0.276 | 0.217 | 0.314 | 0.263 | 0.373 |
| ETTh1 | 96 | **0.485** | **0.450** | 0.500 | 0.465 | 0.498 | 0.476 | 0.512 | 0.490 | 0.512 | 0.490 | 0.520 | 0.499 |
| ETTh2 | 96 | 0.244 | 0.319 | **0.234** | **0.309** | 0.239 | 0.312 | 0.239 | 0.312 | 0.237 | 0.313 | 0.242 | 0.320 |
| ETTm1 | 96 | 0.463 | 0.439 | 0.345 | 0.375 | 0.314 | 0.360 | 0.309 | 0.360 | **0.306** | **0.356** | 0.308 | 0.357 |
| ETTm2 | 96 | 0.141 | 0.250 | 0.134 | 0.242 | 0.131 | 0.239 | **0.125** | **0.234** | 0.129 | 0.237 | 0.126 | **0.234** |

## B.5 PAST DEPENDENCY GRID SEARCH

On the 8 reference datasets presented in Section 3.2, a gridsearch was run to set the best past dependency length for the proposed approach. For each dataset, several input sizes were tested from half of the seasonally to 8 times the seasonality. The best one was selected based on the MSE of the resulting model on the validation set. Table 12 summarizes the results of each gridsearch.

## B.6 EXAMPLE OF PREDICTIONS

Finally, we provide examples of predictions on the 8 reference datasets. Figure 9 and 10 display for each dataset a prediction of the proposed approach along with the prediction of the emission distributions when the hidden state is fixed to 0, 1 or 2.

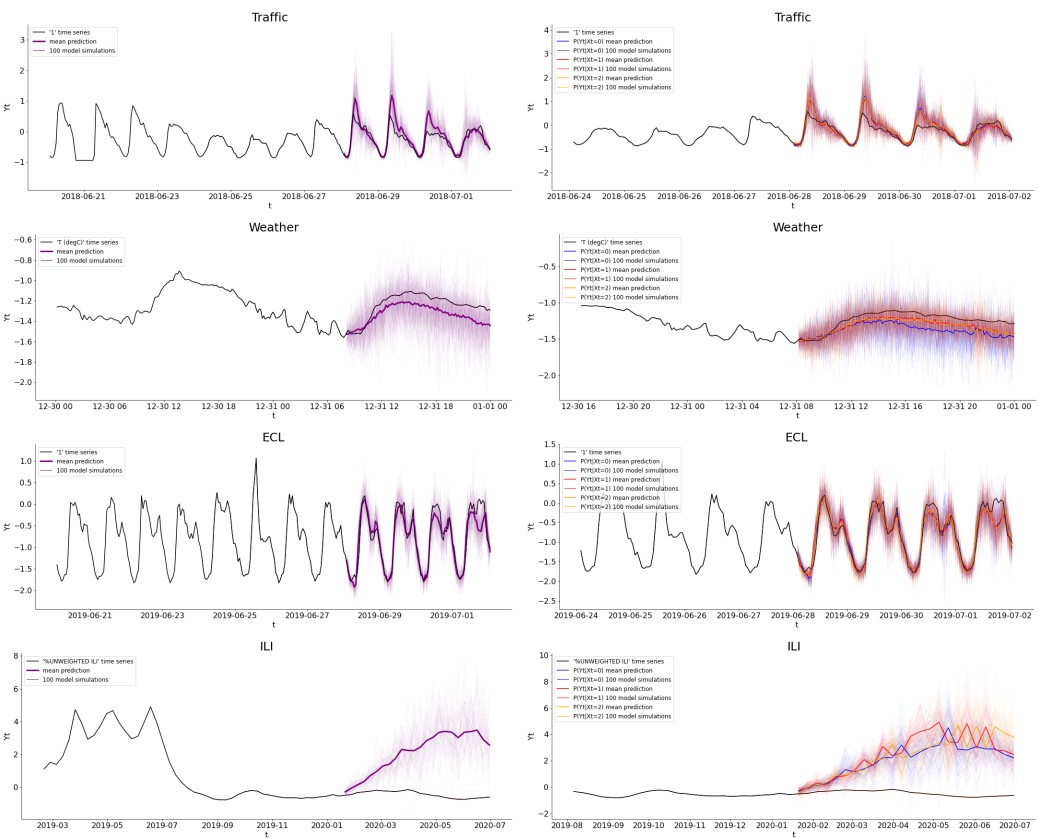

Figure 9: **Predictions on reference datasets.** Example of final prediction of the proposed model on the reference datasets Traffic, Weather, ECL and ILI. (Left) 100 simulations along with the mean prediction of the model (Right) 100 simulations and mean prediction of the three emission distributions when the hidden state is fixed to 0, 1 or 2.

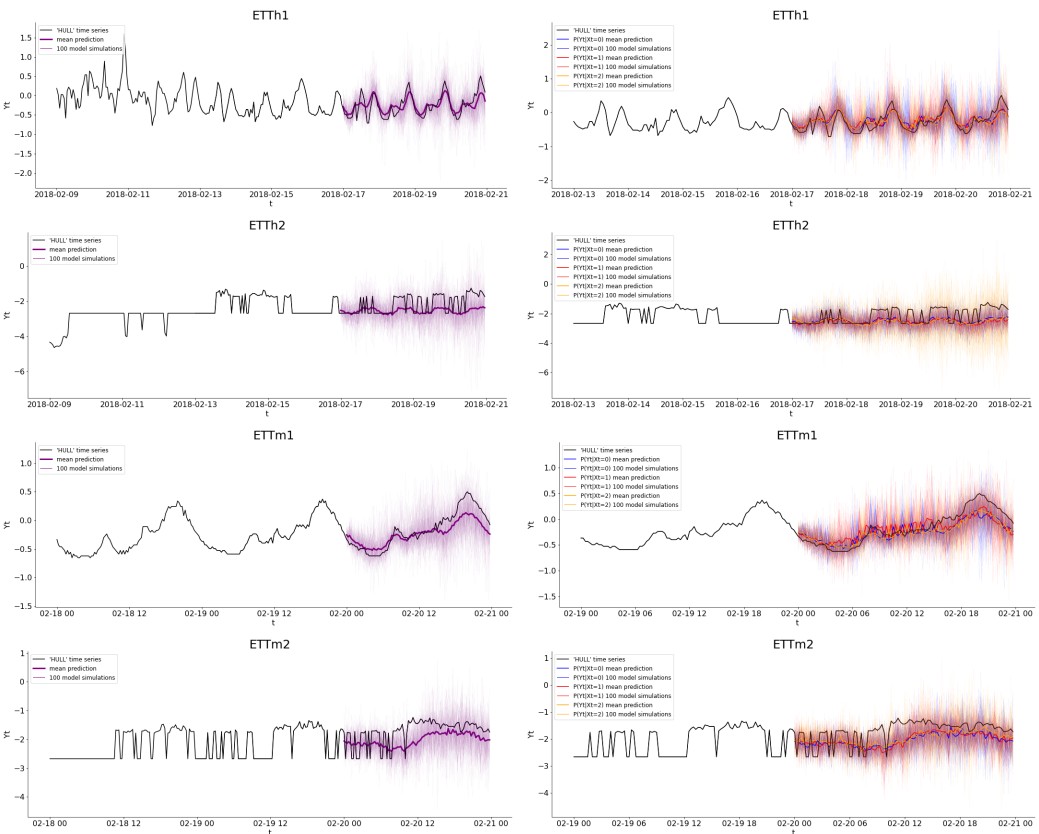

Figure 10: **Predictions on reference datasets.** Example of final prediction of the proposed model on the reference datasets ETTh1, ETTh2, ETTm1 and ETTm2. (Left) 100 simulations along with the mean prediction of the model (Right) 100 simulations and mean prediction of the three emission distributions when the hidden state is fixed to 0, 1 or 2.

