# OpenReview forum: "Variational quantization for state space models"
_ICLR.cc/2024/Conference — Submitted to ICLR 2024_

### Official Review · Reviewer_pvUL · 2023-10-26

**Soundness:** 3 good
**Presentation:** 3 good
**Contribution:** 3 good
**Rating:** 6
**Confidence:** 3

**Summary:**

This paper proposes a new forecasting method by combining hidden Markov models and recurrent neural networks. The training procedure is inspired by vector quantized variational autoencoders, including the latent space and the emission laws parts. This method is computationally efficient and outperforms the SOTA baseline methods.

**Strengths:**

- The presentation of this paper is good. The authors show the comprehensive details of training and network architectures.

- The experiments and evaluation are convincing. The authors test on multiple stationary and non-stationary datasets.

**Weaknesses:**

The writing of this paper can be improved. On Page 2, there is a relatively large blank space, which can be optimized. Also, there are many typos and grammatical issues in this paper. Some of the issues are listed below.
- On page 6, Section 3.1.3, “...where T stand for…”, “stand” should be “stands”.
- On Page 8, in the caption of Table 1, “...their associated MASE…”, “MASE” should be “MASEs”.
- On Page 8, Section 3.2.2, the second equation should be MAE.
- On Page 8, Section 3.2.3, “the accuracy of our model reaches state-of-the-art standards and provide uncertainty quantification.”, the subject of "provide" is not the accuracy.

**Questions:**

In Figure 1, for Hidden States Trajectory, e.g., $\hat{x}_{t+1}^i=2$, are those “2,2,1,1,3,...,3” fixed or flexible to adjust in your implementation?

---

> ### Author Response · Authors · 2023-11-22
> **Response to Reviewer**
>
> We thank the referee for his appreciation of our work and we apologize for these mistakes. They were corrected and we improved the layout of our paper. Regarding Figure 1, the trajectory of hidden states displayed in the figure is an example of a possible sequence of hidden states. Thus, this trajectory is not fixed but sampled using the laws of hidden states. A sentence has been added to the figure description to clarify this point.

---

### Official Review · Reviewer_YnSQ · 2023-10-31

**Soundness:** 3 good
**Presentation:** 3 good
**Contribution:** 3 good
**Rating:** 6
**Confidence:** 3

**Summary:**

This paper proposes a forecasting model that combines discrete state space hidden Markov models with recurrent neural networks. The model is trained in a similar way as VQ-VAE. Experiments on several datasets show that the proposed method outperforms other SOTA methods.

**Strengths:**

(1) Introducing finite state HMM for forecasting is interesting.

(2) The formulation of the model and the learning method is sound.

(3) The experiment results are good.

**Weaknesses:**

(1) The two-stage training appears unnecessary in the context of time series. It may lead to sub-optimal results.

(2) More details should be provided on training the model with discrete latents in the VAE framework, i.e., how the discreteness is handled.

(3) There is only one baseline model based on Transformer. I suspect there are many variants for forecasting.

**Questions:**

(1) How well does your method work on long context forecasting problem?

(2) Do you employ straight-through trick or Gumbel trick in training your model?

---

> ### Author Response · Authors · 2023-11-22
> **Response to Reviewer**
>
> We thank the referee for his appreciation of our work, aligned with the other reviews that we received. You can find below an overview of the major modifications of the manuscript followed by detailed responses to your questions.
>
> - Q1: The two-stage training appears unnecessary in the context of time series. It may lead to sub-optimal results. Do you employ straight-   through trick or Gumbel trick in training your model
>
> A1: Thank you for this thoughtful question. In this paper we propose an efficient, simple and stable way of training our method by training alternatively the emission laws and the prior law of the hidden states. To motivate the proposed two-steps training, we added a justification part in the main text (Section 2.2). Furthermore, a section in Appendix was also added where we compare a one-step training with the proposing one, pointing out instabilities solved by the two-steps training. We acknowledge that additional fine tuning of the implementation could be proposed to improve the training process. This question remains open and will be tackled in the future as well as research perspectives raised in the conclusion of our work.
>
> - Q2 :More details should be provided on training the model with discrete latents in the VAE framework, i.e., how the discreteness is handled.
>
> A2: For a better clarity, several ablation studies were added in the appendix in which we questioned some of the choices we made during our implementation. For example, we review in Appendix A.5 the importance of model size, in Appendix A.6, the impact of freezing or learning the variance of our emission laws and in Appendix A.7, the improvements made when we let hidden states depend on time.
>
> - Q3:There is only one baseline model based on Transformer. I suspect there are many variants for forecasting.
>
> A3: We thank the reviewer for this constructive comment. We added in Table 1 two other Transformer-based methods (Informer and TimesNet).
>
> - Q4: How well does your method work on long context forecasting problem?
>
> A4: In addition to the experiments conducted on the fashion dataset (with a horizon set to 52), we also evaluated our method on a collection of 8 benchmark datasets in Section 3.2. The main motivation for this second set of experiments was to evaluate our approach on different forecasting use cases and evaluate its behavior over different forecasting horizons. We found that the proposed approach can be easily used in a wide range of forecasting tasks while providing accurate forecasts that challenge other state-of-the-art methods.

---

### Official Review · Reviewer_9MfL · 2023-10-31

**Soundness:** 3 good
**Presentation:** 3 good
**Contribution:** 3 good
**Rating:** 6
**Confidence:** 3

**Summary:**

The authors present a new algorithm for time-series prediction based on VQ-VAEs and RNNs, with separate emission models for each quantized hidden state. The authors evaluate their algorithm on a number of datasets, showing comparable results with SOTA algorithms.

**Strengths:**

* The authors evaluated their algorithm against multiple other algorithms on multiple datasets, with multi-seed comparison + grid search.
* As far as I am aware, combining VQ-VAEs and RNNs in this specific way (latent-conditioned observation model) has not been explored before.

**Weaknesses:**

* While the emprical results are nice, it would further strengthen the paper if the authors could perform ablation experiments to uncover/provide a better intuition about why their algorithm performs well against others.

**Questions:**

* The author's new algorithm performs almost similarly to previous SOTA Transformer-based model PatchTST/64. However, it's unclear from my first reading why one would prefer one over the other. Is it easier to train/better runtime etc.?
* It's nice that the authors performed a grid search over learning rates and batch size for the other algorithms. I think the paper would be further strengthened if the authors conducted a hyperparameter search also over network size where it make sense (similar to the grid search that the authors performed over hidden size for their network)?

---

> ### Author Response · Authors · 2023-11-22
> **Response to Reviewer**
>
> We thank the referee for his appreciation of our work, aligned with the other reviews that we received. You can find below an overview of the major modifications of the manuscript followed by detailed responses to your questions.
>
> - Q1: While the emprical results are nice, it would further strengthen the paper if the authors could perform ablation experiments to uncover/provide a better intuition about why their algorithm performs well against others.
>
> A1: We thank the referee for this suggestion. Following this remark, several ablation studies were added in the appendix. For instance, in Appendix A.6, we provided more justification on the two-steps training process presented in the paper, showing how it solves some training instabilities and improves final model accuracy. Likewise, we reviewed in Appendix A.7 the impact of learning the variance of the emission distributions and in Appendix A.8, the impact of using a Markov chain for the latent states.
>
> - Q2:The author's new algorithm performs almost similarly to previous SOTA Transformer-based model PatchTST/64. However, it's unclear from my first reading why one would prefer one over the other. Is it easier to train/better runtime etc.?
>
> A2: We thank the referee for this thoughtful remark and recognize that the motivation for the method presented was not clearly defined enough in the paper. One argument in favor of our method is that it provides probabilistic predictions that allow users to build confidence intervals. On the Fashion use case, by analysing when and where the emission laws having access to the influencers signals is activated, we could spot specific regime shifts in the time series and understand how the proposed approach succeeded at forecasting these challenging patterns. A natural future work would be to propose more complex generative models where Transformer architectures could be used in the transition densities and emission distributions. Such models would benefit from the state space framework and highlight the usefulness of the two stage procedures.
>
> - Q3: It's nice that the authors performed a grid search over learning rates and batch size for the other algorithms. I think the paper would be further strengthened if the authors conducted a hyperparameter search also over network size where it make sense (similar to the grid search that the authors performed over hidden size for their network)?
>
> A3: We thank the referee for this suggestion. Thanks to this comment, we provided a complementary ablation study presented in appendix A.5. In this section, we performed a grid search on the number of parameters and compared the optimal size found for our approach with the size of other state-of-the-art methods. We found that our model performed well with an architecture with 700,000 parameters, which is superior to PatchTST, Informer, and DeepAR methods and inferior to N-HiTS, N-BEATS, and TimesNet methods. Considering recent published models for natural language processing or visual detections, there is still a huge margin regarding the size of all current time series forecasting methods.

---

### Meta-Review · Area_Chair_jXFY · 2023-12-11

**Metareview:**

The aim of this paper is to develop a latent variable time series model. The model is setup as a HMM. The emission function comprises K neural networks each of which characterizes a Gaussian distribution. The prior distribution over the hidden states is parameterized by a function (neural network) that outputs a matrix. The learning algorithm involves first learning the emission function and the variational approximation followed by the transition function (keeping the remaining parameters fixed).

At the end of the review period this paper was a borderline submission. While the empirical evaluation of the method was sound, I think the paper still has significant room for improvements in terms of the clarity of writing and exposition (particularly with respect to what is new for the community that works on deep generative models for time series data).

It is unclear what the biggest contributions of this work to the community are. Right now the paper reads as a combination of neural networks, state space models and a curriculum style learning algorithm. Its not immediately made obvious to the reader what parts are new and what is not.

For example there is a long line of research in latent variable deep generative modeling using recurrent neural networks combined with latent variables (https://arxiv.org/abs/1506.02216), Gaussian state space models (https://arxiv.org/abs/1609.09869), and models of recurrent switching dynamics (https://papers.nips.cc/paper/2020/hash/aa1f5f73327ba40d47ebce155e785aaf-Abstract.html). Beyond contextualizing these (and their follow up) works I think its important to give the reader an understand of what axes the proposed approach differs on, why its better and ideally some form of empirical comparison to the same.

There is also prior related work on the idea of blending tools from VQ-VAEs with time-series modeling: e.g. see

* https://arxiv.org/abs/2205.15894
* https://arxiv.org/abs/2303.04743

If indeed the main contribution is that these precise combination of ingredients result in a model that outperforms recent works on forecasting then its important to atleast compare with variants of deep generative models for time series data that exist in the literature to understand and unpack what facets of this work lend itself to improved results.

Finally some minor points for further improvements to clarity please also (a) number equations, (b) ensure Figures and legends and ticks are legible (e.g. Fig 2's legend is not legible unless one zooms in significantly) and (c) please expand on the parametric choices made for the variational approximation in the implementation section.

Overall I do think the work itself is interesting and encourage the authors to refine the submission further.

**Justification For Why Not Higher Score:**

Justification provided in main review.

**Justification For Why Not Lower Score:**

N/A

---

### Decision · Program_Chairs · 2024-01-16

Reject